# Transcription regulates the spatio-temporal dynamics of genes through micro-compartmentalization

Hossein Salari[1,2] ✉, Geneviève Fourel[1] & Daniel Jost [ID][1] ✉

Although our understanding of the involvement of heterochromatin architectural factors in shaping nuclear organization is improving, there is still ongoing debate regarding the role of active genes in this process. In this study, we utilize publicly-available Micro-C data from mouse embryonic stem cells to investigate the relationship between gene transcription and 3D gene folding. Our analysis uncovers a nonmonotonic - globally positive - correlation between intragenic contact density and Pol II occupancy, independent of cohesin-based loop extrusion. Through the development of a biophysical model integrating the role of transcription dynamics within a polymer model of chromosome organization, we demonstrate that Pol II-mediated attractive interactions with limited valency between transcribed regions yield quantitative predictions consistent with chromosome-conformation-capture and live-imaging experiments. Our work provides compelling evidence that transcriptional activity shapes the 4D genome through Pol II-mediated micro-compartmentalization.

Chromosome conformation assays like Hi-C unveiled hierarchical organization of chromosomes within eukaryotic nuclei[1,2]. In metazoans, Mbp-scale "checkerboard" patterns in contact maps reveal spatial segregation of chromosomes into a euchromatic "A" compartment and a heterochromatic "B" compartment[3,4]. At a smaller scale (~100 s kbp), chromosomes fold into topologically associating domains (TADs) and loops[5–7]. The prevailing model[8] suggests that compartments emerge from the micro-phase separation of epigenomic regions mediated by chromatin-binding architectural proteins[9,10], while most TADs result from cohesin-driven chromatin loop extrusion with CTCF acting as a barrier[11,12]. Hi-C and live imaging experiments indicate that depleting CTCF or cohesin disrupts TADs, weakens CTCF-mediated loops, but has limited effects on compartmentalization[7,13–16]. Nevertheless, some loops or TADs remain unaffected by these treatments[14,17], likely originating from distinct mechanisms.

3D chromosome organization regulates gene expression during interphase[18–20]. Notably, colocalization of promoters and enhancers within TADs can directly influence transcription initiation, potentially increasing transcription rates[20,21]. Conversely, recent studies suggest that genes serve as central units of the 3D genome and that transcription itself plays a role[22–25]. High-resolution contact maps in mammalian and fly cells reveal transcription-dependent fine structures, such as loops between active gene promoters, promoters and enhancers, or transcriptional start (TSS) and termination (TTS) sites of the same gene[13,26–29]. However, the mechanistic origins of these fine structures, despite their potential significance in gene regulation, remain controversial.

Indeed, on the one hand, some experiments in Drosophila and mice indicate higher 3D contacts within expressed gene bodies compared to repressed ones[29,30]. Remodeling of chromatin structure around genes during mouse thymocyte maturation often coincides with transcriptional changes[31]. RNA Polymerase IIs (Pol II) form also distinct foci and higher-order clusters known as transcription factories[32–35], and active genes tend to colocalize within transcriptionally active subcompartments[27,36]. On the other hand, there are cases in mammalian cells where significant unfolding of genes occurs

[1]Laboratoire de Biologie et Modélisation de la Cellule, École Normale Supérieure de Lyon, CNRS, UMR5239, Inserm U1293, Université Claude Bernard Lyon 1, 46 Allée d'Italie, 69007 Lyon, France. [2]École Normale Supérieure de Lyon, CNRS, Laboratoire de Physique, 46 Allée d'Italie, 69007 Lyon, France. ✉e-mail: hossein.salari@ens-lyon.fr; daniel.jost@ens-lyon.fr

after strong transcriptional activation[37–39], and acute depletion of Pol I, II, and III has minimal effects on large-scale genome folding[40]. In budding yeast, gene activity inversely correlates with local chromatin compaction[25]. Live-cell imaging experiments highlight the relationship between gene transcription and chromatin dynamics[41–45], revealing enhanced gene mobility upon Pol II elongation inhibition[41,43,44] or gene activation[42] and correlated motions between active regions[44,45].

Complementing experiments, biophysical and polymer models have also explored the complex interplay between transcription and genome dynamics[35,37,46–49]. Elongating or backtracked Pol IIs may act as barriers for SMC-mediated extrusion, indirectly impacting genome organization[38,46,47,50–54]. Transcription-dependent changes in the local chromatin fiber rigidity and contour length may lead to extended conformations for highly transcribed genes[37]. Computational models considering Pol II-mediated interactions[35], interactions with transcription factories or condensates[48,49], or P-TEFb interactions[43] suggest that attractive interactions between active regions may capture intergene contacts observed in Hi-C and the gene mobility observed in live-imaging.

Overall, the evidence presents a complex understanding of the genome spatio-temporal dynamics in response to gene transcription, necessitating a comprehensive framework to reconcile these observations. In this study, we analyze publicly available Micro-C data for mouse embryonic stem cells (mESCs) and develop observables to characterize transcription-dependent 3D gene folding. Our analysis reveals a nonmonotonic relationship between intragenic contact density and gene transcription, potentially reconciling contradictory data. By dissecting the contributions of loop extrusion and transcription-associated factors, we propose Pol II occupancy as a key determinant of gene folding. Using a traffic model for gene activity and a 3D polymer model[55,56], we demonstrate that transcriptionally active subcompartments and intragenic contact enrichment may arise from Pol II-mediated phase separation. Furthermore, we suggest that Pol II-mediated condensation, coupled with transcriptional bursting, may slow down gene mobility, aligning with experimental observations.

## Results

### RNA Pol II occupancy and gene length correlate with intra-gene condensation

In this study, we aimed to investigate the potential role of transcriptional activity in the local organization of genes within the genome. To accomplish this, we focused on mouse embryonic stem cells (mESC) as they provide abundant quantitative data. Specifically, we utilized publicly available high-resolution Micro-C and Pol II ChIP-seq data[13]. For our analysis, Micro-C contact maps were distance-normalized to examine contact enrichment compared to a sequence-averaged null behavior, resulting in the observed over expected (obs/exp) contact map. We introduced two scores for each gene (Fig. 1A and Methods): (i) Intra-gene contact enrichment (IC), which represents the mean obs/exp values calculated for all pairs of loci within the gene, capturing the level of self-association and overall gene condensation. (ii) Intra-gene RNA Pol II enrichment (IR), which corresponds to the mean normalized Pol II ChIP-seq profile within the gene and reflects gene transcriptional activity, correlating with RNA-seq data (Supplementary Fig. 1).

Fig. 1B shows a significant positive correlation (Spearman's $\rho = 0.56$, $t$ test $p$-value < 1e-200) between IC and IR scores, indicating that increased transcriptional activity is associated with enhanced intra-gene condensation. This result is consistent with prior research on mouse ESC[29], *Drosophila*[30] and mouse DP and DN3 thymocytes[31]. We also checked that such a correlation remains mainly independent of the phosphorylation status (Ser5P and Ser2P) of Pol II (Supplementary Fig. 2), which is associated with different dynamical and

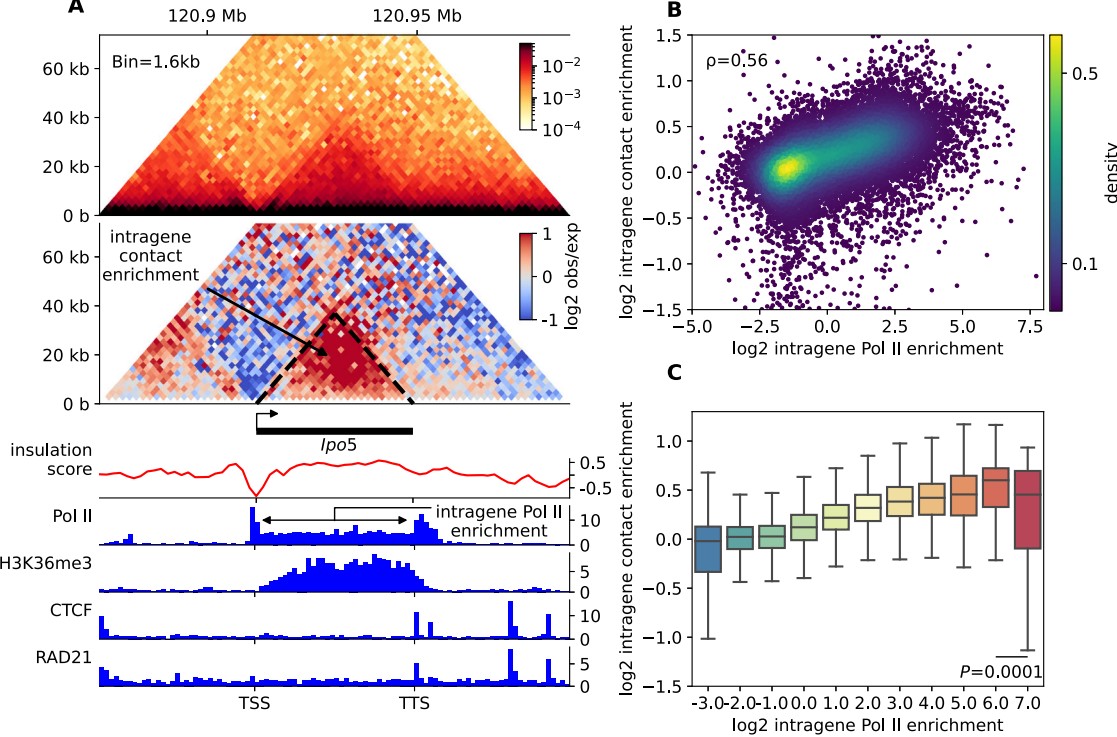

**Fig. 1 | Correlation between intra-gene contact and intra-gene Pol II enrichments. A** Observed Micro-C contact map (top), observed/expected map (middle) and several ChIP-seq profiles (bottom) of the genomic region including the *Ipo5* gene (chr14:120,874-120,984 kb) in mESC. The intra-gene contact (IC) and RNA Pol II (IR) enrichments are illustrated. **B** Scatterplot of IC versus IR for all genes longer than 1 kb (24,363 genes). Colors refer to the density of dots. **C** Boxplots of IC after clustering together the genes (dots in (**B**)) with similar IR. The number of genes in each cluster from left (IR score = −3) to right (IR score = 7) are, respectively, 175, 1629, 6603, 4515, 3926, 3673, 2463, 946, 265, 109, and 37. Boxplots present the median and 25th and 75th percentile, with the whiskers extending to 1.5 times the interquartile range. Two-tailed t-tests were performed between the two last clusters 6 and 7, $p$-value = 0.0001. Source data are provided as a Source Data file.

interacting states of the polymerase[57]. Additionally, a similar correlation between IC and intra-gene H3K36me3 (a histone mark related to Pol II elongation) content was detected (Supplementary Fig. 1). As a control, we observed weak, negative correlations between IC and repressive marks (H3K27me3 and H3K9me3) (Supplementary Fig. 3). Clustering genes based on similar IR scores revealed a nonlinear and non-monotonic relationship (Fig. 1C): IC generally increases with IR, except at very high Pol II levels where a slight but significant relative decrease in contact frequency occurs. Importantly, this behavior cannot be solely attributed to the inherent properties of Micro-C experiments to detect more or less contacts depending on the molecular crowding on DNA[25] since similar behavior was observed using mESC Hi-C data[19] (Supplementary Fig. 4). Interestingly, this correlation between IR and IC holds true regardless of gene compartment (A or B) (Supplementary Fig. 5) or the number of exons[19] (Supplementary Fig. 6). However, genes with higher exon counts tend to exhibit more intra-gene contacts compared to those with fewer exons. Moreover, IC scores for A-compartment genes are generally higher than those for B-compartment genes, which may suggest an interference between the segregation of heterochromatin and the condensation of Pol II-enriched genes. Interestingly, this difference becomes more pronounced for genes with high IR scores, where the drop in IC is more significant for B-genes.

In mammals, highly active genes are typically smaller, and larger genes, when active, are usually lowly expressed (Supplementary Fig. 7). Hence, we investigated whether gene size may be a confounding factor or, on the contrary, could be a determining factor by classifying genes based on both their IR and genomic length (Methods). Figure 2B displays the average IC score for each category, revealing a positive correlation between IC and gene length: longer genes exhibit stronger intra-gene contact frequency at a given Pol II occupancy density. These findings suggest a cooperative effect in intra-gene folding, wherein both Pol II density and gene size play integral roles[58].

To gain deeper insights into the contact patterns and profiles within and surrounding genes, we conducted a pile-up meta-gene analysis (PMGA), aggregating the rescaled obs/exp maps and ChIP-seq profiles of genes with similar size and transcriptional activity (Fig. 2A, C and Methods). PMGA uncovered a strong correlation between Pol II profiles and certain structural features of contact maps: intra-gene contact maps were nearly uniform, consistent with the constant Pol II levels observed within genes; stripes of preferential interactions were observed between Pol II-rich promoters/TSSs and gene bodies (stripe); and loops were formed between Pol II-rich TSSs and TTSs (TSS-TTS loops). Notably, the correlation between Ser2P Poll II (which have no peak at TSS), Ser5P Pol II (only weak peaks at TSS and TTS) and H3K36me3 (strong depletion around TSSs) profiles, taken individually, and Micro-C patterns such as TSS-TTS loops and stripes was less apparent (Fig. 2A, C and Supplementary Fig. 2). Regarding the dependency on gene size, we found that promoters of short genes are often located at the domain borders, while larger genes tend to form their own insulated domains separate from surrounding regions.

To further investigate the role of Pol II occupancy, we analyzed two publicly available datasets involving the treatment of mESC cells with transcriptional inhibitor drugs: triptolide (TRP), which inhibits Pol II initiation, and flavopiridol (FLV), which inhibits Pol II elongation[29]. Firstly, we confirmed the significant reduction in the intensity of Pol II-mediated loops after both treatments (Supplementary Fig. 8). Consistent with the observed loss of intra-gene Pol II occupancy in all genes, particularly highly transcribed ones (Supplementary Fig. 9 and Supplementary Fig. 10), intra-gene interactions were weaker in the TRP and FLV cases compared to the normal condition (Fig. 3A, Supplementary Fig. 11 and Supplementary Fig. 12), resulting in a 12% reduction in IC for large active genes post-treatment. These results align with a previously reported observation of 25% reduction in the intensities of gene stripes following Pol II inhibition[29]. Moreover, there exists a

notable correlation between the fold-changes (treated vs untreated) in IC and IR scores (Fig. 3C): the greater the reduction in Pol II occupancy for a given gene, the more likely its intra-gene folding is affected. Interestingly, when re-clustering genes based on their new IR scores measured in TRP- and FLV-treated cells, we still observed an average increase in IC as a function of IR similar to the untreated case (Fig. 3D), suggesting that the remaining intra-gene interactions observed after transcription inhibition may be attributed to residual Pol II occupancies. This 'master curve' provides further evidence that Pol II level only is predictive—in average—of the intra-gene folding whatever the conditions (treated or untreated).

In summary, our findings demonstrate a correlation between intra-gene condensation, interaction patterns, local transcriptional activity, gene length, and Pol II occupancy.

## Cohesin-mediated loop-extrusion activity plays a minor role on intra-gene condensation

Recent studies, both experimental and theoretical, have proposed that the loop extrusion mechanism, which plays a crucial role in the formation of TADs, might have an impact on the transcription machinery[23,38,46,47,54]. Interestingly, we observed a significant correlation between the occupancy of CTCF and cohesin, the main players in loop extrusion, and the intra-gene contact enrichment and Pol II occupancy (Fig. 2C and Supplementary Fig. 1). This observation led us to investigate whether cohesin-mediated loop extrusion could drive the correlation between transcriptional activity and intra-gene folding discussed earlier.

To address this question, we analyzed our original dataset from wild-type mESCs and excluded genes with high SMC1a (a cohesin subunit) occupancy (Methods). The remaining genes, clustered based on IC and gene length, showed significantly lower levels of CTCF and cohesin, while Pol II profiles remained largely unchanged (Supplementary Fig. 13C). Despite this subset of genes, the IC and IR scores still exhibited a strong correlation at a level similar to wild-type (Fig. 3D, Supplementary Fig. 13A, B). Additionally, PMGA revealed that the typical interaction patterns observed within genes were still visible for cohesin-poor genes, although certain features such as stripes, which are known to be footprints of loop extrusion activity near extruding barriers[38,46], were absent outside the genes (black arrows in Fig. 3B left).

Furthermore, we utilized three publicly available mESC datasets where CTCF (ΔCTCF), the cohesin subunit RAD21 (ΔRAD21), or the cohesin unloader WAPL (ΔWAPL) were acutely depleted[13]. These treatments led to significant alterations in CTCF and cohesin occupancies throughout the genome[13], as well as changes in TAD folding and CTCF-CTCF loops[7,14], such as a strong reduction in loop intensity in ΔCTCF and ΔRAD21 and reinforcement in ΔWAPL (Supplementary Fig. 8 bottom). However, most gene expressions remained unaltered[13], and the majority of loops between Pol II peaks were unaffected (Supplementary Fig. 8 top). Surprisingly, despite the acute changes in intra-gene CTCF and cohesin profiles (Supplementary Fig. 14C, Supplementary Fig. 15C and Supplementary Fig. 16C), we observed only minimal effects on intra-gene interactions (Figs. 3B, 3D, Supplementary Fig. 12, Supplementary Figs. 14–16A, B). The most noticeable—yet weak—changes in IC scores occurred in highly active genes (high IR), with an average 5% reduction in ΔRAD21 (Fig. 3B). However, the changes in IC between WT and ΔRAD21 conditions did not exhibit a clear correlation with changes in RAD21 occupancy (Supplementary Fig. 17). Similar to cohesin-poor genes in WT, the structural features associated with loop extrusion outside genes were lost or significantly reduced in ΔCTCF and ΔRAD21 (and enhanced in ΔWAPL) (Fig. 3E).

Collectively, these results indicate that cohesin-mediated loop extrusion does not significantly affect the specific organization of transcribed genes, suggesting the presence of an independent mechanism.

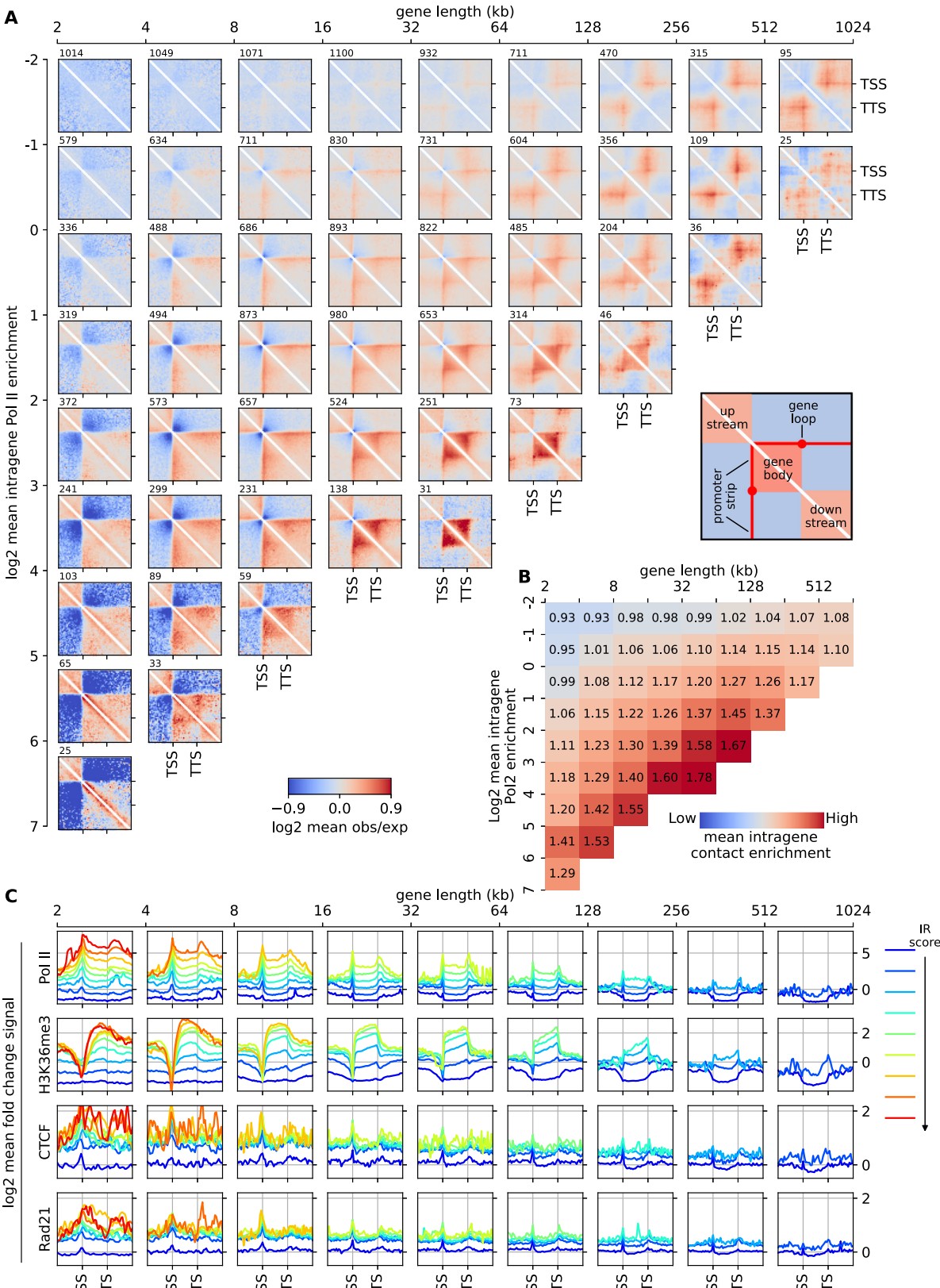

**Fig. 2 | Gene classification based on size and Pol II occupancy, and pileup meta-gene analysis. A** Pileup meta-gene analysis (PMGA, see Methods) of the obs/exp map around genes clustered based on their length (horizontal axis) and Pol II enrichment (vertical axis). The number of genes of each cluster is indicated above on each map. Maps for clusters with less than 25 representative genes were not drawn, due to lack of statistics. **B** Average IC scores for each cluster in (**A**). **C** PMGA of different chromatin tracks: in each subplot, all the average profiles of the different Pol II clusters for genes of the same length range are shown (from left to right: from small to large genes); different colors correspond to the different Pol II clusters, from low (blue) to high (red) IR score (respectively, −2,−1,0,1,2,3,4,5,6). Source data are provided as a Source Data file.

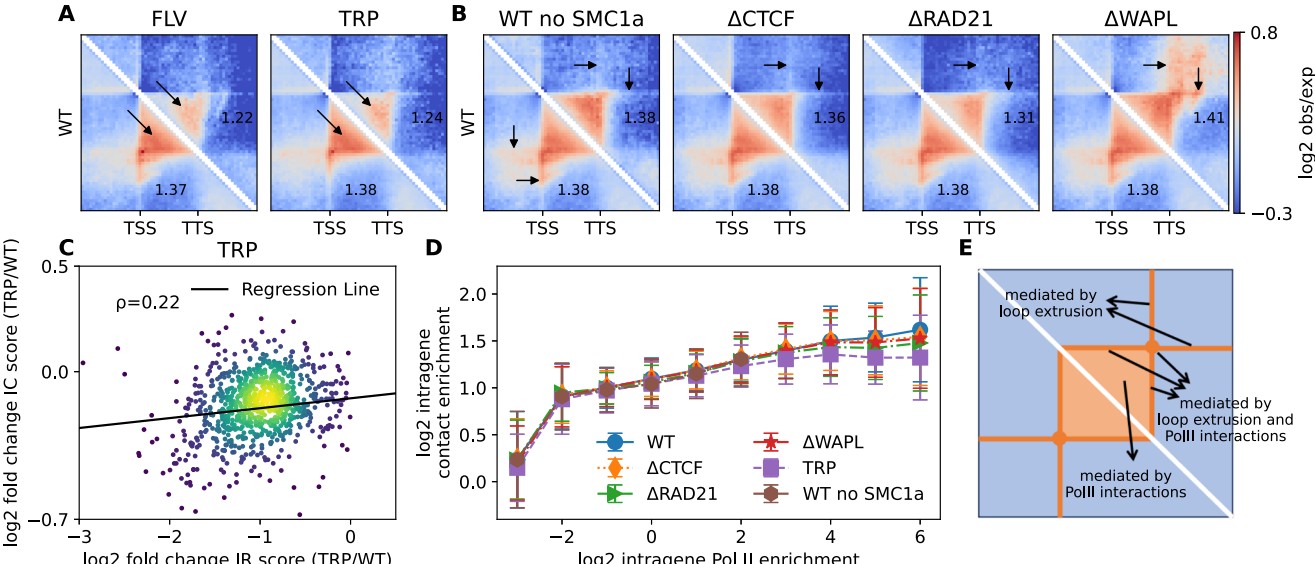

**Fig. 3 | Gene conformation is affected by acute change in Pol II but not in CTCF and cohesin. A** Comparison between PMGA of untreated WT cells and cells treated with transcription inhibitors for genes with size of 64-128 kb, $IR_{wt} > 1$ in WT condition and with a reduced IR score in treated cells ($IR_{treat.} < IR_{wt}$). **B** PMGA for genes with size of 64-128 kb and $1 < IR_{wt} < 2$ in conditions of reduced CTCF, RAD21 or WAPL levels or for a subset of genes with low SMC1a level in WT cells (WT no SMC1a, most left). **C** Scatter Plot of fold change of intragene contact enrichment against the fold change in Pol II occupancy after TRP treatment for the genes >64 kb, $IR_{wt} > 1$ in WT condition and with a reduced IR score in treated cells ($IR_{treat.} < IR_{wt}$). The Spearman correlation is given. **D** The intragene contact enrichment upon acute depletion of RAD21, CTCF and WAPL, by IAA treatment of an engineered ES cell line, or by treatment with triptolide (TRP), as a function of IR score in the treated cells. The Spearman's correlation between average IC and IR scores of WT is 0.97. Data are presented as mean values ± SD and were computed over a number of genes always higher than 10 (median number = 1735). **E** Scheme summarizing the different determinants of structures observed inside or around active genes. Source data are provided as a Source Data file.

## A biophysical model to investigate the role of transcription on gene folding

Our data analysis strongly suggests that Pol II occupancy drives the 3D organization of genes, independently of cohesin activity. Moreover, recent in vitro and in vivo experiments suggest that Pol IIs could form liquid-like droplets either directly through a phase-separation process mediated by weak interactions between their carboxy-terminal domains[36,59–62] or indirectly via the formation of Mediator condensates triggered by nascent RNAs[63]. In the following, we developed a biophysical model to better characterize the phenomenology of Pol II-mediated gene folding by investigating how effective self-attractions between Pol II-occupied loci may shape the spatio-temporal dynamics of genes.

First, we built a stochastic model to describe Pol II occupancy and dynamics at a gene using a standard Totally Asymmetric Simple Exclusion Process (TASEP)[64–66]. In this model (Fig. 4A, Methods), Pol IIs can be loaded onto chromatin at the TSS with rate, transcription elongation initiates with rate $\gamma_0$, Pol IIs then progress along the gene at rate until they unbind from chromatin at TTS with a rate. During this process, Pol IIs cannot overlap or bypass each other. We systematically varied the parameters of the TASEP model in order to predict different Pol II profiles along the gene at steady-state (Supplementary Fig. 18). For example, by varying model parameters ($\gamma_0/\gamma = 1$, $\beta/\gamma = 1 - \alpha/\gamma$), we reproduced uniform average profiles of Pol II occupancy along the gene, ranging from low (~0.02) to high (~0.80) densities (Fig. 4B, C).

Next, to assess the spatial organization of a gene, we integrated the TASEP in a 3D polymer model of chromatin fiber[55,56] (Fig. 4A, Methods). Briefly, we represented a 20 Mbp-long section of chromatin as a self-avoiding chain (1 monomer = 2kbp = 50 nm). We focused on a region of size $L$ in the middle of the chain, which represents the gene of interest. Each monomer within the gene is characterized by a random binary variable indicating the local Pol II occupancy, whose dynamics is described by the TASEP. To investigate the impact of Pol IIs density and dynamics on gene folding, we assumed that monomers occupied by Pol II at a given time may self-interact at short-range with energy strength $E$. All the other monomers are considered non-interacting, neutral particles. The coupled stochastic spatio-temporal dynamics of the Pol II occupancies and 3D positions of the monomers are then simulated using kinetic Monte Carlo (Methods).

## Self-attraction between Pol II-bound genomic regions drives the intra-gene spatial organization

We quantified generic structural properties of the model and investigated the relationship between intra-gene condensation (IC scores) and Pol II density (IR scores) with respect to model parameters. In particular, we varied IR scores (via $\alpha/\gamma$) while keeping other TASEP parameters constant, achieving uniform Pol II occupancies along the gene (as in Fig.4B, C), and we monitored the corresponding IC scores at steady-state.

For fixed gene length $L$ and elongation rate $\gamma$, IC is an increasing function of both the Pol II density (Fig. 4D, upper panels) and the strength $E$ of self-attraction (Fig.4E, mid panel): the gene's polymeric subchain undergoes a theta-like collapse[67] towards a globular state when the Pol II occupancy reaches a critical value (Supplementary Movie 1), such transition occurring at lower threshold densities for stronger interactions ($|E|$). Similar to standard self-interacting homopolymers[68,69], intra-gene contacts strengthen with increasing gene length (Fig.4F, left panel), while maintaining a fixed average Pol II level. This reflects the cooperative nature of the theta-collapse[70,71].

At a constant average Pol II density, IC is a decreasing function of the Pol II elongation rate (Fig. 4E, upper panel). Indeed, the capacity of Pol II-bound monomers to stably interact depends on the out-of-equilibrium dynamics of the elongating Pol IIs: shorter residence time of Pol II on a monomer (compared to typical polymer diffusion time) results in more transient Pol II-mediated interactions between monomers. Notably, biologically relevant elongation rates (~2 kb/min[72],) correspond to the slow elongation regime, maximizing gene condensation.

Overall, our model qualitatively recapitulates the global Pol II and gene length trends observed experimentally (Fig.2B).

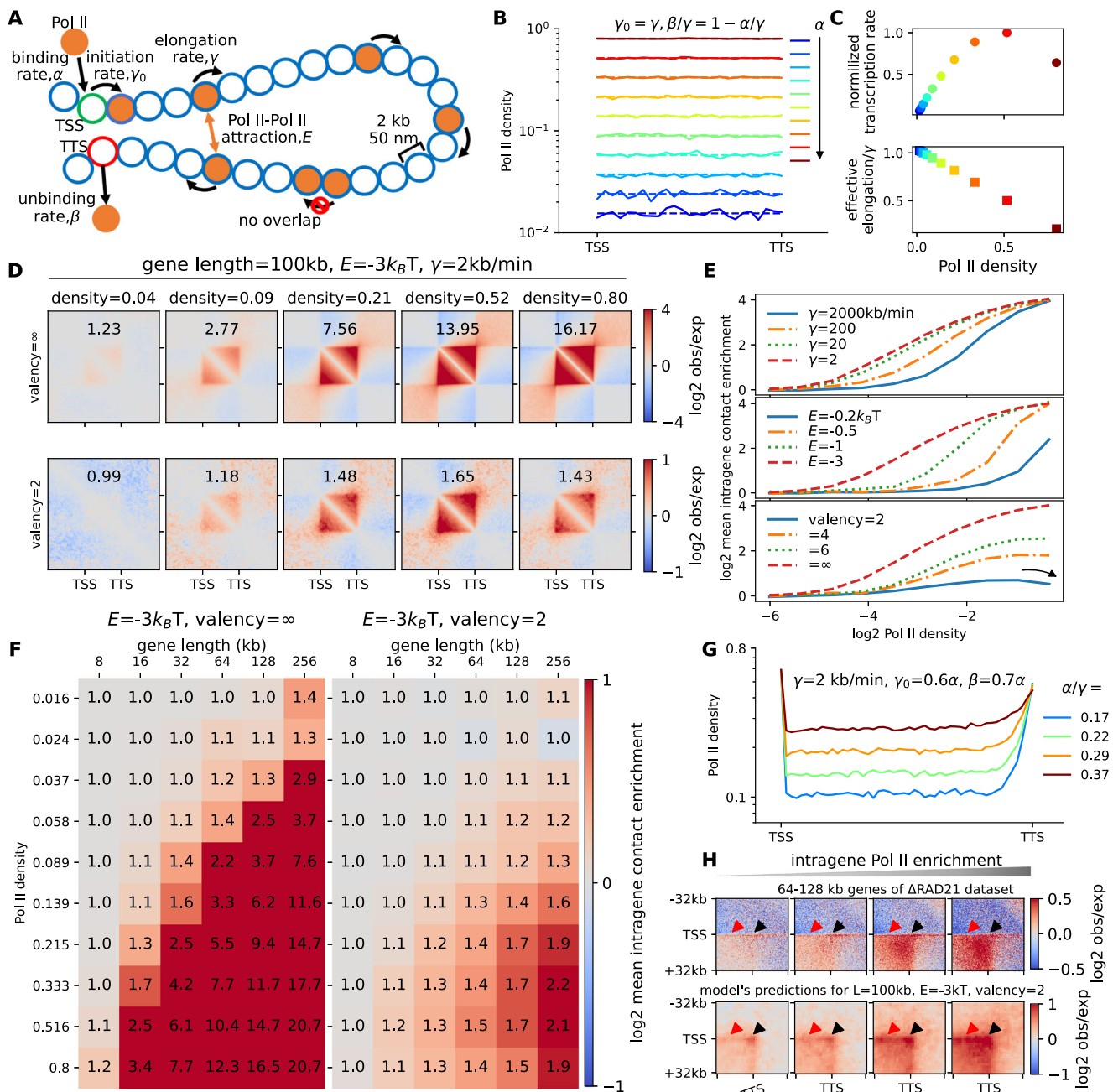

**Fig. 4 | Transcription-mediated interactions regulate gene folding. A** Schematic representation of the TASEP-decorated polymer model for gene transcription and 3D folding. **B** Pol II profiles along a 100kbp-long gene ($L = 50$ monomers) for parameters tuned to generate a uniform occupancy along the gene, from low (blue) to high (red) densities, (respectively, 0.016, 0.024, 0.037, 0.058, 0.089, 0.139, 0.215, 0.333, 0.516, and 0.800). The solid and dashed curves are predictions from Monte-Carlo simulations and analytical calculations, respectively (see Methods). **C** Normalized transcription rate (top), defined as the average number of Pol II unloadings from the TTS per time unit divided by its maximum, and normalized effective elongation rate (bottom), defined as the inverse of the time needed for one Pol II to fully transcribed a gene, as a function of Pol II density. **D** Predicted contact maps around a 100kbp-long gene for different Pol II densities and valencies. Corresponding IC scores are given. **E** IC versus IR curves as a function of the elongation rate $\gamma$ (top), strength of interaction $E$ (middle) and valency (bottom). **F** IC scores against IR scores (Pol II density) and gene length for two different valencies. The color bar is presented in a log2 scale, while the values are given in a linear scale. **G** Examples of non-uniform Pol II profiles having significant accumulations at TSS and TTS. **(H) (Top)** PMGA analysis of the contact around the TSS-TTS loop for 64–128 kb-long genes with increasing IR scores (from left to right) taken from $\Delta$RAD21 dataset. **(Bottom)** Model predictions around the TSS-TTS loop for the non-uniform cases described in (G). TTS-TTS loops and promoter stripes are shown with black and red arrows, respectively. Source data are provided as a Source Data file.

However, the predicted strengths of intra-gene contact enrichment are much stronger than expected (Fig. 4F, left panel). For instance, a 128kbp-long gene shows a ~6-fold increase in IC score with a ~8-fold rise in Pol II density across the theta-collapse (for $\gamma = 2$ kb/min and $E = -3$ kT), whereas experimentally the same change in average Pol II occupancy yields only ~35% increase in IC. We verified that reducing $|E|$ does not resolve the problem as the

theta-transition remains sharp and cooperative (Supplementary Fig. 19, left panel).

### Intra-gene condensation in mESC is consistent with a limited valency of Pol II-Pol II interactions
In our initial model, unrestricted interactions were allowed among the Pol II-occupied monomers in close proximity in the 3D space.

However, such molecular interactions are mediated by only a restricted set of accessible residues and thus one monomer may have only a limited valency (number of simultaneous interactions).

Reducing the valency led to a global, sharp drop in intra-gene contact enrichment (Fig. 4D, 4E, lower panels, Supplementary Movie 2). For instance, at high Pol II occupancy, a ~11-fold reduction in IC score was observed for valency 2 compared to unlimited valency. At lower valencies (2 or 3), the levels of contact enrichment aligned with experimental values (Fig. 4F, right panel, Supplementary Fig. 19, right panel) while still preserving the overall dependence on Pol II density and gene length seen with unlimited valency.

However, an intriguing exception emerged: the IC score now displays a non-monotonic dependency with Pol II levels (Fig. 4E, lower panel), as actually observed experimentally at high IR scores (Fig.1C). Within our framework, this behavior arises from a screening effect on long-range interactions. At high Pol II density, the neighboring Pol II-occupied monomers along the chain are likely to engage in interactions, limiting the ability of a monomer to interact with distantly located monomers and consequently reducing large-scale intra-gene condensation.

## Nonuniform Pol II profiles lead to intra-gene architectural details

We previously focused on average gene folding properties by considering flat, homogeneous Pol II densities. However, experimental Pol II profiles show distinct peaks at TSS and TTS. By adjusting the TASEP parameters, we generated qualitatively similar peaked profiles of increasing density (Fig. 4G, Methods). Using interacting parameters ($E = -3kT$, valency = 2) compatible with the experimental IC vs IR relationship, we obtain for these nonuniform profiles very similar correlations between IC and IR scores and gene length (Supplementary Fig. 20). Additionally, we predicted the formation of a stable loop between TSS and TTS in contact map as well as promoter-gene stripes within gene body, for high Pol II occupancy (Fig. 4H, lower panels). Interestingly, off-diagonal pileup analysis of mESC Micro-C datasets around TSS-TTS anchors exhibits similar patterns independent of the cohesin loop-extrusion mechanism (Fig. 4H, upper panels, Supplementary Fig. 21), implying that such architectural details are driven by Pol II occupancy and effective Pol II-Pol II interactions.

## Stochastic dynamics of gene folding in response to transcription bursting

Most mammalian genes undergo discontinuous transcription in bursts[73–75]. To address the impact of such bursting kinetics on the gene spatio-temporal dynamics, we modified the TASEP model minimally: the promoter can stochastically switch between an on state, enabling Pol II binding and transcription, and an off-state refractory to Pol II binding, with rates $k_{on}$ and $k_{off}$ (Fig. 5A). These rates define the effective Pol II binding rate ($\alpha_{eff} = \alpha k_{on}/(k_{on} + k_{off})$), the burst frequency ($= k_{on}k_{off}/(k_{on} + k_{off})$, mean number of bursts per time unit) and the train size ($= \alpha/k_{off}$, mean number of Pol II binding and elongating during one burst). For simplicity, we assumed $k_{on} = k_{off} \equiv k$, allowing variation in burst properties from rare, long trains ($k = 0.01$/min) to frequent, short ones ($k = 0.04$/min) (Fig. 5B, C), while maintaining an almost constant average Pol II density profile (Fig. 5D).

By averaging over all configurations, we observed a weak−but significant−decrease in intra-gene condensation in the presence of bursting (Fig. 5E). However, when considering the promoter's on/off states separately, the impact of bursting became apparent with overall more intra-gene contacts and more pronounced TSS-TTS loops and promoter-gene stripes in the on-state (Fig. 5G). This effect was more pronounced for low burst frequency as the difference in Pol II occupancy between the on/off-states became more prominent (Fig. 5F). Similarly, more elongating trains lead to increased condensation (Supplementary Fig. 22).

These findings suggest a time-correlation between transcriptional bursting and gene folding where dynamical changes in the gene's radius of gyration (RG) are preceded by modifications in Pol II along the gene (Fig. 5H). Indeed, we observed an overall negative correlation between instantaneous Pol II density and RG, which was more pronounced for low burst frequencies (Fig. 5I, J). Interestingly, when multiple trains are present simultaneously along the gene, the dynamic looping between could rise to the formation of 'factories' where they colocalize (Fig. 5K, Supplementary Movie 3).

## Transcription slows down gene mobility

Live-imaging experiments have indicated that chromatin motion is enhanced after Pol II inhibition or reduced after gene activation[41,43,44], suggesting a connection between transcription and a reduced gene mobility. To assess whether our biophysical model aligns with these observations, we computed for each monomer the mean-squared displacement (MSD), that measures the typical space explored by a locus over a time-lag Δt. We observed that $MSD \sim D\Delta t^{\delta}$, where $D$ and $\delta$ are diffusion constant and exponent, respectively (Fig. 6A). $\delta \sim 0.5$ is independent of Pol II occupancy (Fig. 6B) and its value is consistent with live imaging experiments in mESC[15,16] and standard polymer dynamics[76,77]. Conversely, $D$ depends on Pol II density and gene length (Fig. 6C) with a perfect opposite trend as the intra-gene condensation (Fig. 4F, right): the more condensed the gene the less mobile[77]. For example, a 40-70% increase in intra-gene contacts corresponds to a 10-15% decrease for $D$, consistent with experiments (Fig. 6C, D).

## Transcription-associated long-range contacts correlate with Pol II occupancy

Our analysis of intra-gene folding and dynamics suggests that similar mechanisms may explain the role of Pol II occupancy in distal inter-gene interactions. On the Micro-C map of mESC, we observed selective contact enrichments between distal highly active genes (Fig. 7, Supplementary Fig. 23). For instance, the average contact frequency between the 811 kb-distant large active genes *Ahctf1* and *Parp1* is 3.2-fold higher than expected at similar genomic distance (Fig. 7A). Both genes belong to the same A compartment, indicating that strongly transcribed genes may further colocalize within A. To test this hypothesis, we clustered all the 32-64 kb-long genes into three categories based on their IR score (Low, Mid and High) and performed PMGA (Methods) of the inter-gene contacts for pairs of genes distant by more than 128 kb but less than 2 Mb (Fig. 7B, Supplementary Fig. 24). When both genes are transcribed (Mid and High clusters in Fig. 7B), a strong promoter-promoter interaction is detected, as already observed in several studies[13,27,29,78]. In addition, PMGA highlights that highly active genes (High-High) also exhibit significant contact enrichment between their gene bodies compared to the surrounding background in a transcription-dependent and loop extrusion-independent manner (Supplementary Fig. 24). Contact enrichment between inactive genes (Low-Low) is similar to background and can be attributed to their location in the more compact B-compartment[30].

To rationalize these observations with our biophysical model, we conducted simulations for two 60 kbp-long genes distant by 460 kb, exhibiting similar steady-state Pol II profiles (Fig. 7C). We observed that Pol II-mediated interactions not only affect intra-gene contacts but also drive the formation of inter-gene contacts between TSS and TTS and between gene bodies, whose strengths increase with Pol II density. We obtained similar results for longer genes and shorter inter-gene distances (Supplementary Fig. 25). Interestingly, interacting genes tend to colocalize and segregate from the rest of the simulated polymeric chain[79,80] (Fig. 7D, Supplementary Movie 4).

## Discussion

In this study, we analyzed publicly available Micro-C data of mESC[13,29] to investigate the relationship between transcriptional activity and

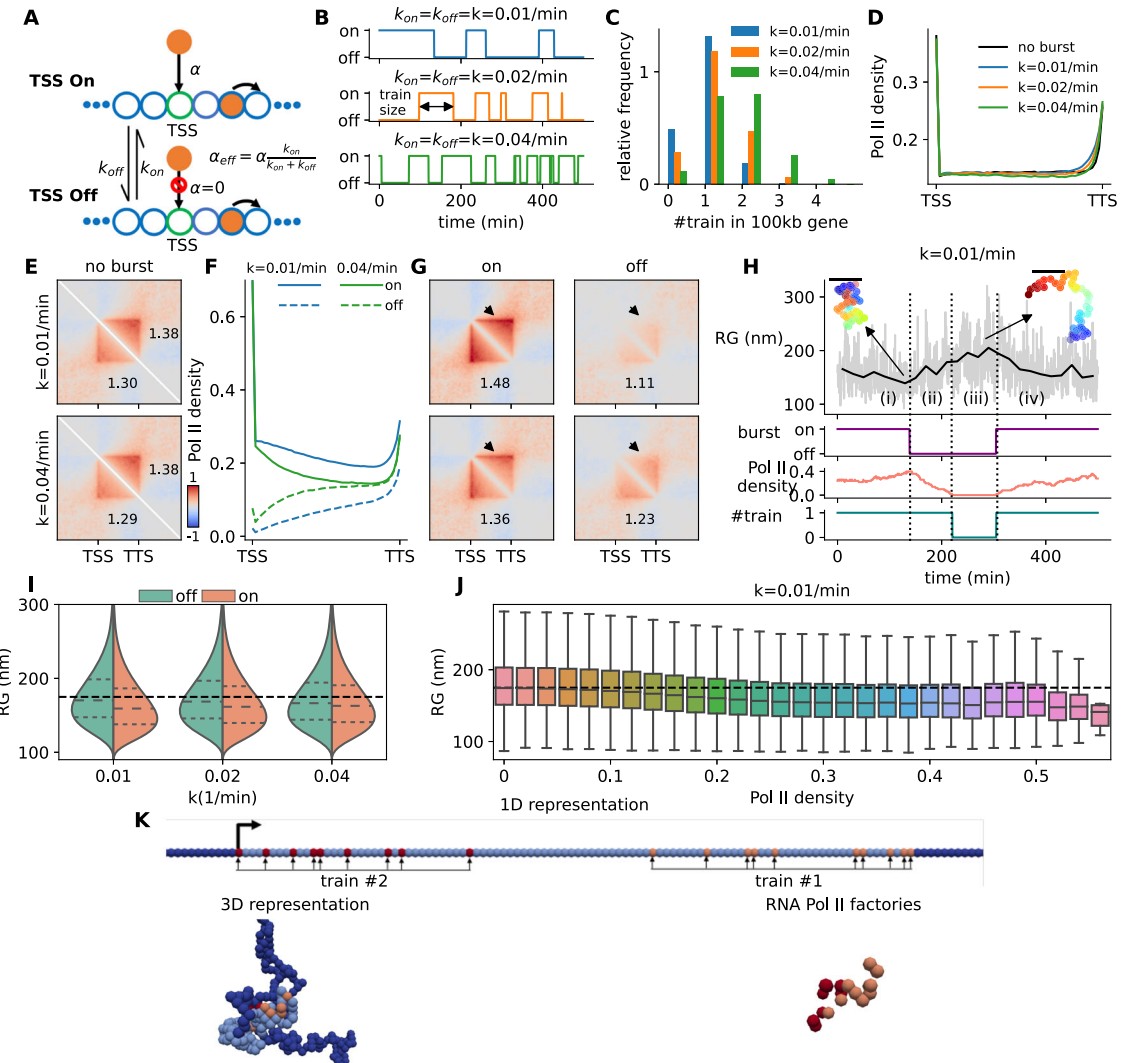

**Fig. 5 | Transcriptional bursting leads to dynamical changes in gene conformation. A** Schematic representation of transcriptional burst, where TSS alternatively switched on and off. **B** Three different examples of bursty gene activity ranging from long ($k = 0.01$/min) to short ($k = 0.04$/min) train size. **C** Probability distributions of the number of trains elongating on a gene at the same time for the three bursty regimes depicted in (**B**). **D** Average Pol II density profiles for the three bursty regimes depicted in (**B**) and in the absence of burst. **E** Predicted contact maps with (lower left triangular part) and without burst (upper right triangular part) for long (top) and short (bottom) trains. **F** Pol II density profiles when TSS is "on" (solid lines) or "off" (dashed lines) for long (blue lines) and short (green lines) trains. **G** Predicted contact maps for conditions similar to (**F**). Color scale is the same as panel (**E**). **H** (Top to bottom) Time evolution of the radius of gyration (RG) of a gene, TSS state, Pol II density along the gene and the number of trains

elongating along the gene for $k = 0.01$/min. Examples of 3D gene conformation are drawn when the gene is more or less condensed. Bars = 200 nm. **I** Violin plots of RG in the "off" and "on" states for the three burst regimes in (**B**). The black dashed lines show the predictions for homopolymer model (i.e. zero interaction case). **J** Boxplot of RG as a function of the Pol II density for $k = 0.01$/min. Boxplots present the median and 25th and 75th percentile, with the whiskers extending to 1.5 times the interquartile range. They were computed over a number of snapshots always higher than 10 (median number ~$10^5$). **K** A typical snapshot of gene 3D conformation (gene in light blue, flanking regions in dark blue) in the presence of two trains. The 1D representation shows the locations of Pol II-bound monomers for each train (orange and red dots). All simulations were done for a 100-kb gene with valency = 2, $E = -3k_BT$. Source data are provided as a Source Data file.

chromosome organization. Our findings align notably with previous studies[29–31]. Specifically, we showed that, on average, at the single-gene level (2kbp-1Mbp), intra-gene contact enrichment, structural patterns (gene-loops, promoter-stripes, Fig. 3E) and the degree of insulation from the surrounding genomic regions correlate positively with Pol II occupancy along the gene (Figs.1,2). Moreover, our study revealed that these observed features also exhibit a positive correlation with gene length (Fig.2), suggesting a cooperative mechanism for gene folding. Nevertheless, we noted a considerable degree of heterogeneity, implying that specific genes may deviate from the average behavior. These results stand in contrast with the very local structure of the chromatin fiber (<600 bp) that is increasingly open as transcription rate increases[25].

For highly expressed genes, we observed reduced contacts within gene body (Fig. 1C), which aligns, although at a lesser extent, with the extended gene conformations observed for very long, highly expressed tissue-specific genes in mice[37,39].

Consistent with prior research[47], our results underscore the role of the loop extrusion process, recognized for driving the formation of loops and TADs[11] and reported to interfere with transcriptional elongation[38,46,47], into structural features outside the gene domain (Fig. 3). However, in good agreement with recent high-precision Capture Micro-C data[27], we demonstrated that intragenic structure-function relation between gene condensation and gene transcription does not directly associate with loop extrusion[79] (Fig. 3). At the inter-gene level, we observed long-range contacts between active genes, not only

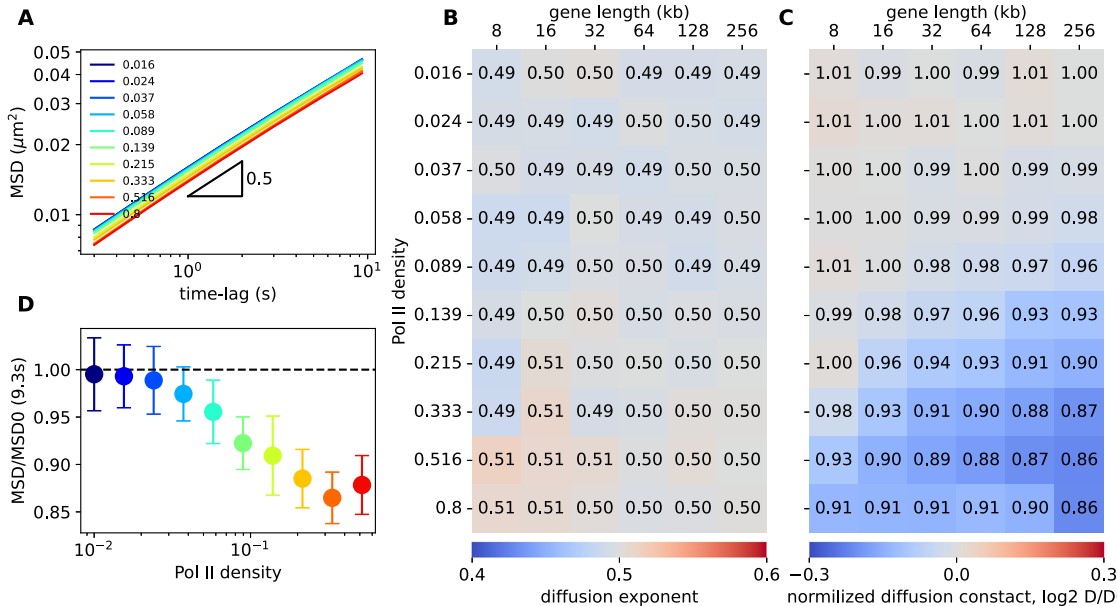

**Fig. 6 | Transcription activity slows down gene mobility. A** Mean-squared displacement $MSD \sim D\Delta t^\delta$ vs time-lag $\Delta t$ for different Pol II densities for a 256kbp-long gene. **B** Diffusion exponent $\delta$ as a function of gene size and Pol II density. **C** As in (**B**) but for the diffusion constant $D$ normalized by its value $D_O$ in the absence of transcription. The color bar is presented in a log2 scale, while the values are given in a linear scale. **D** The ratio of MSD with ($MSD$) and without ($MSDO$) Pol II at t = 9.3 s as a function of Pol II density for a 256kbp-long gene. Color scale as in (**A**) and data are presented as mean values ± SD and were computed over 20 different trajectories. All simulations were done with valency = 2, $E = -3k_BT$. Source data are provided as a Source Data file.

between gene promoters as already characterized[81,82], but also between gene bodies (Fig. 7), here also closely tied to Pol II profiles and independent of loop extrusion (Supplementary Fig. 24).

Altogether, our findings suggest that active genes are central units of the 3D genome[25] and form a subcompartment[27,79,80], driven by gene activity, Pol II binding and elongation. This observation likely holds true for other cell types, as we recently showed that intra-gene folding during mouse thymocyte maturation is, in average, also associated with change in transcription levels[31]. The mechanisms described here are also likely to be broadly conserved in animals. Indeed, we analyzed the correlation between IC and IR (spearman's ρ = 0.48) in whole-embryo *Drosophila* data at embryonic nuclear cycle 14 (Supplementary Fig. 26)[83]. *Drosophila* is interesting as its chromosome organization is believed to be mainly driven by the spatial segregation of the epigenome instead of cohesin loop-extrusion processes[4]. We found a similar nonmonotonic dependence of IC to IR as well as Pol II-related intra-gene interaction patterns. One exception is the effect of gene length that is less clear. Interestingly, in the bacterium *Escherichia coli*, higher transcription is also associated with more intra-gene contacts[84]; in yeast and dinoflagellate, TAD-like structures are associated with (blocks of) active genes[25,85].

To better characterize the underlying mechanisms behind the correlations between Pol II activity and the transcriptionally active subcompartment, we introduced a simple biophysical framework that accounts for the 1D dynamics of Pol II along genes coupled to the 3D polymer organization of chromosomes (Fig. 4). Previous biophysical models have already addressed some aspects of Pol II-mediated phase separation via attractive[35,48] or active forces[86], focusing on large-scale inter-gene condensation, but never investigating intra-gene organization nor explicitly accounting for the transcription dynamics. By assuming self-attractive, short-range interactions between genomic loci bound to Pol II[35,48], our approach is able to recapitulate qualitatively the overall augmentation of intra-gene contacts associated with an enrichment of Pol II density inside gene body and to longer genes, consistent with a standard cooperative coil-globule transition observed for finite-size chains[71,87–89]. Our model suggests that limiting

the number of possible interactions per Pol II-bound region to low values (e.g., 2 or 3) allows us to align quantitatively our predictions with experiments, leading to percolated but less condensed 3D domains[90,91]. Interestingly, this constraint also explains the weak decompaction observed for highly transcribed genes as interactions between distant positions along the genes (mediating the large-scale gene's condensation) are screened by (more frequent) interactions between nearest-neighbor Pol II-bound sites. This screening mechanism may also contribute to the formation of the extended transcription loops observed in long highly transcribed genes[37], along with the potential stiffening of the chromatin fiber caused by the high density of nascent ribonucleoprotein complexes along the genes, as originally evoked.

Furthermore, our model predicts a strong coupling between gene structure and dynamics: transcription bursts may regulate the stochasticity of intra- and inter-gene contacts at the single-cell scale (Fig. 5)[92]; such dynamical contacts may conversely reduce locally gene mobility (Fig. 6) and lead to long-range coherent motion between active regions[56,93], in good agreement with live-microscopy observations[41,43,45].

What are the molecular determinants of the putative attractive interaction between Pol II-bound loci hypothesized in our model? It is likely that several sources may directly or effectively participate in its regulation. The C-terminal domain (CTD) of Pol II can form liquid condensates in vitro under physiological conditions, which become unstable upon CTD phosphorylation[59]. This mechanism may thus promote direct attractions in vivo between non-elongating Pol II, bound at promoters for example[35]. CTDs may also interact with co-factors that can themselves phase-separate both at the transcriptional initiation[34,94,95] and elongation[60,96] stages, like FUS, BRD4, Mediator, P-TEFb or splicing factors. For example, the observed correlation between intra-gene condensation and the number of exons[19] at similar Pol II occupancy (Supplementary Fig. 4) suggests a role for splicing-related condensates[96]. In addition, transcription-generated supercoiling[84,97] or specific histone marks deposited along the

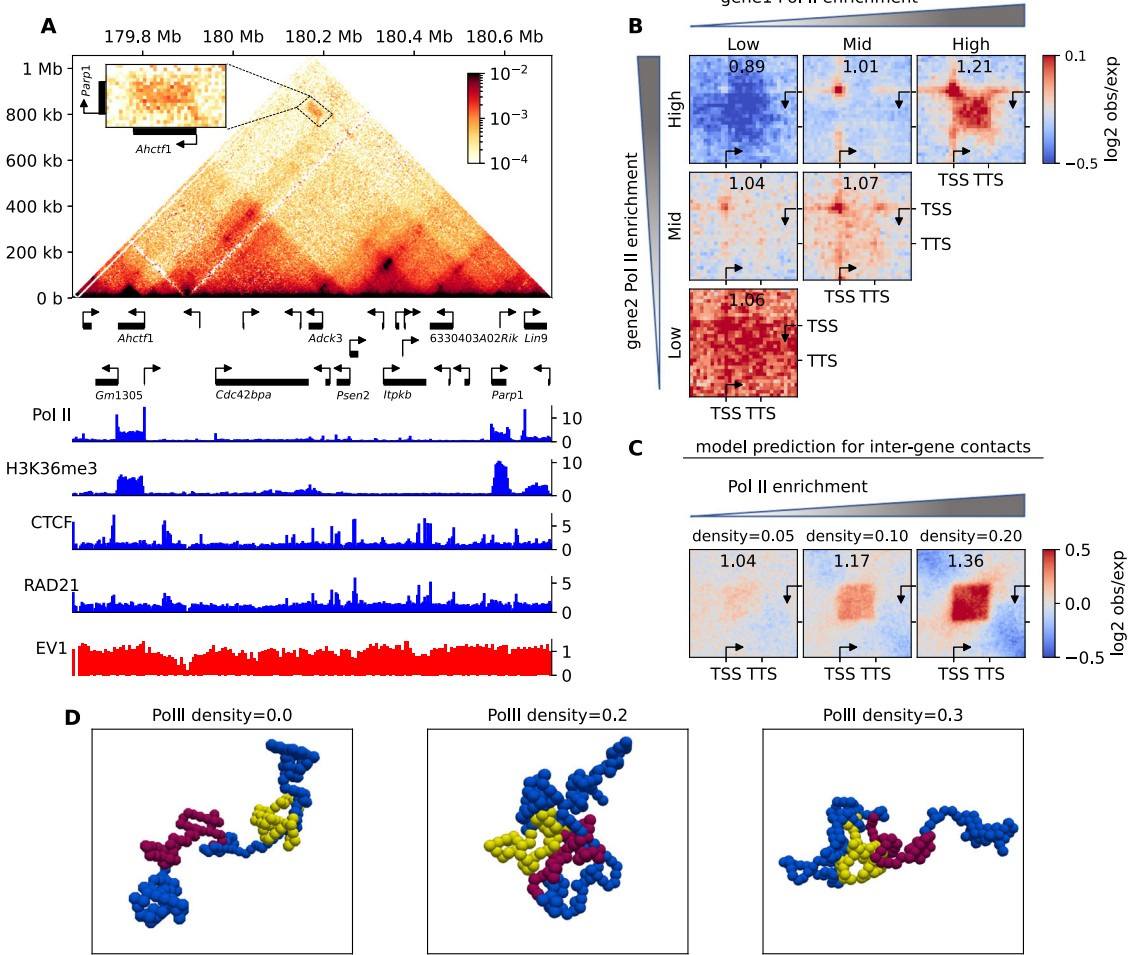

**Fig. 7 | Inter-gene contacts between active genes. A** Micro-C contact map of a ~1 Mb region of mESC chromosome 1, with corresponding gene annotation and ChIP-seq profiles below. Inset shows a zoom between the long, highly-active genes of *Ahctf1* and *Parp1* (respectively, 58.7 kb and 32.3 kb-long and an expression of 22.4 FPKM and 151.4 FPKM). **B** Inter-gene pileup meta-gene analysis of the contact enrichment between two distant genes as a function of their intra-gene Pol II enrichment. **C** Model predictions for contacts between 60-kb-long genes for three different Pol II densities. **D** Examples of simulated 3D configurations illustrating the inter-gene interactions at various Pol II densities (gene regions in yellow and red, surrounding genomic regions in blue). All simulations were done with valency=2, $E = -3k_BT$. Source data are provided as a Source Data file.

gene bodies (that may regulate putative nucleosome-nucleosome interactions[98]) may contribute to transcription-dependent effective interactions.

The limited valency of interactions in our model aligns with a restricted number of simultaneously accessible residues involved in the aforementioned sources of Pol II-Pol II attraction. It is also possible that the screening effect observed at high transcription rates could be explained by the strength of interaction depending on local Pol II concentration and/or the length of nascent transcripts (Supplementary Notes), as RNA size and concentration can impact the stability of transcription-related condensates[63].

In conclusion, our results demonstrate the significant impact of Pol II binding and elongation on the spatiotemporal organization of the active genome through an out-of-equilibrium phase-separation process coupling the time-dependent dynamics of transcription to the formation of gene micro-domains and of transcriptionally active subcompartment[27,61,79,99]. This extends the concept of transcription factories[100], typically associated with inter-gene contacts, to the internal organization of long genes having multiple trains of transcribing Pol IIs. Consistent with our findings, recent works also proposed that interactions between Pol IIs may also facilitate promoter-enhancer communications[23,101]. However, our approach provides

only an "average" picture of the role of transcription on 3D chromosome organization and does not account for the various epigenetic, genomic and spatial factors that may interplay with Pol II-mediated phase separation[47] around specific genes, potentially explaining the variability of behaviors observed after transcription (de)activation[31].

Future investigations should aim to further elucidate the biological function(s) of such transcription-dependent micro-compartmentalization. Indeed, colocalization of active genomic regions may enhance the recycling of Pol II or transcription co-factors[102,103] by increasing their local concentrations. Investigating precisely such a "structure-function" coupling between the binding and assembly of transcription-associated components and condensates and the spatial folding of the genome remains an intriguing challenge and would require further developments both at the experimental and modeling levels.

## Methods
### Experimental data analysis
**Datasets.** The processed Micro-C data for mESCs (wild-type and mutants) and *Drosophila* in multi-resolution format mcool were downloaded from National Center for Biotechnology Information's

Gene Expression Omnibus (GEO) through accession no: GSE130275, GSE178982 and ArrayExpress accession E-MTAB-9306.

The ChIP-seq tracks, including Pol II, Pol II Ser5P and 2P, CTCF, RAD21, H3K27me3, H3K9me3 and H3K36me3, for wild type and different mutants in BigWig format were downloaded from GEO through accession no: GSE130275, GSE178982, GSE90893, GSE90994, GSE16013, GSE85191, GSE195830.

## Pileup meta-gene analysis (PMGA)

**Contact maps.** We used cooltools (https://github.com/open2c/cooltools)[104] module to compute the obs/exp maps from the balanced contact maps, at various resolutions ranging from 100 bp to 50 kb.

To perform intra-gene PMGA, for each gene $i$ with size $l_i$ (>20 x resolution), we considered a domain of size $3l_i$ around it, including the gene body and the two upstream and downstream flanking regions, each of size $l_i$. To ensure consistency and facilitate pileup analysis, we rescaled each corresponding $(3l_i)x(3l_i)$ obs/exp matrix to a (60,60)-pseudo-sized matrix by averaging the original matrix elements. An example of this rescaling process can be seen in Supplementary Fig. 27. We then aligned all the rescaled matrices in the transcription forward direction to maintain uniformity. Finally, we aggregated all the data of genes belonging to a given cluster (clustered by gene length, IR score, etc.).

For inter-gene PMGA, for each pair of genes, we considered the off-diagonal region of the obs/exp map of size $(3l_1)x(3l_2)$ and centered at $(m_1,m_2)$, with $l_1$ and $m_1$ the size and genomic position of the middle of gene 1 (same for gene 2). Then, similarly, we rescaled this region to a (30,30)-pseudo-matrix, aligned the genes in parallel forward direction and aggregated the pseudo-matrices belonging to the same cluster.

**ChIP-seq tracks.** Using pyBigWig (https://github.com/deeptools/pyBigWig), for each gene, we discretized the $3l$ domain (see above) into 60 bins and computed the coverage for each bin. Then, we aligned the domains in the transcription forward direction and aggregated over all genes in the same cluster.

**ChIP-seq peak calling and calculation of peak contacts.** For each ChIP-seq track, we transformed BigWig to bedGraph, used MACS software[105] version 2 to call the peaks in the "no model" mode and merged the results from different replicates. Then, we sorted them by fold-change score (compared to input) and selected the most significant peaks (top 1/3). Finally, for every pair of peaks with a genomic distance between 160 and 320 kb, we used the off-diagonal pileup module of cooltools to compute the average peak contacts.

**Insulation score and compartments analysis.** For computing the insulation score, we analyzed contact maps at 800-bp, 1600-bp and 3200-bp resolutions with the dedicated module of cooltools with sliding windows 3, 5, 10 and 25 times larger than the given resolution, e.g. 2.4, 4, 8 and 20-kb windows for 800-bp resolution. For the compartment analysis, we used the eigs_cis module of cooltools to compute the first eigenvector of Pearson's correlation matrix of contact map taking as inputs the 6.4-kbp resolution Micro-C maps and the GC coverage computing from mm10 reference genome.

**Intra-gene contact (IC) and Pol II (IR) scores.** For each gene, the IC score is defined as the average value of the obs/exp map ($c_{i,j}$) inside gene body: $IC = \frac{\sum_{i>j} c_{ij}}{N(N-1)/2}$, where $i,j$ both represent bins within the same gene and $N$ is the total number of bins along the gene, given by the gene size divided by the contact map resolution. Similarly, the IR score of a gene is defined as the average value of RNA Pol II ChIP-seq signal ($p_i$) within the gene body: $IR = \frac{\sum_i p_i}{N}$.

## Biophysical model

We previously introduced a self-avoiding semi-flexible polymer model for chromosomes[55,56]. In this study, we employed a coarse-graining approach to represent a 20-Mbp-long chromatin fiber using 10,000 monomers. Each monomer corresponds to approximately 2-kbp of the genome and has a size of 50 nm (Fig. 4A). Within this chain, we inserted a 100-kbp-long gene (composed of 50 monomers), where TSS and TTS are located at the first and last monomers of the gene, respectively.

**TASEP model.** Each monomer $i$ within a gene (of total size $n$) is characterized by a binary state $s_i\epsilon\{0,1\}$ depending if a Pol II complex is bound to it ($s_i=1$) or not ($s_i=0$). We simulated the stochastic dynamics of Pol II binding, unbinding and elongation using a simple kinetic Monte-Carlo framework: each Monte Carlo step (MCS) consisted of (i) one attempt to bind a Pol II with rate $\alpha$ at the TSS if unoccupied ($s_1=0 \to 1$), (ii) one attempt to unbind Pol II with rate $\beta$ at the TTS if occupied ($s_n=1 \to 0$), and (iii) $n-1$ elongation attempts, each consisting in randomly picking one monomer $i$ in $[1:n-1]$ and, if occupied, to move with rate $\gamma$ the Pol II to its adjacent upstream monomer if it is not already occupied ($[s_i=1,s_{i+1}=0] \to [0,1]$).

In a simple case where the initiation rate is equal to the elongation rate (i.e. $\gamma_0=\gamma$), the time-evolution of ensemble-averaged $<s_i>$ of monomer $i$ follows

$$d<s_i>/dt = \gamma(<s_{i-1}(1-s_i)> - <s_i(1-s_{i+1})>) \tag{1}$$

Assuming that the state of each monomer is independent from the states of its neighbors and taking the continuum limit $<s_i>(t) \equiv \rho(x,t)$ with $x=i/n$ leads to the Fokker-Planck-like equation

$$\partial_t\rho(x,t)\approx\gamma n^{-1}\left\{(2n)^{-1}\partial_x^2\rho(x,t)+(2\rho(x,t)-1)\partial_x\rho(x,t)\right\} \tag{2}$$

At steady state ($\partial_t\rho(x,t)=0$), $\rho(x)$ is given by solving the differential equation:

$$(2n)^{-1}\partial_x^2\rho(x)+(2\rho(x)-1)\partial_x\rho(x)=0, \tag{3}$$

with $\rho(0)=\alpha/\gamma$ and $\rho(1)=1-\alpha/\gamma$. For the boundary condition $\alpha/\gamma=1-\alpha/\gamma$, the solution is uniform $\rho=\alpha/\gamma$ along the gene. Equation (3) for other conditions can be solved numerically and compared to the results of Monte-Carlo simulations (Supplementary Fig. 18).

**Polymer model.** The polymer chain undergoes local movements on a FCC lattice with periodic boundary conditions under Metropolis criterion, as described in our previous works[55]. The total Hamiltonian of a given configuration can be expressed as following:

$$H=\kappa\sum_{i=2}^{N-1}(1-\cos\theta_i)+E\sum_{i,j}f_{ij}s_is_j \tag{4}$$

The first term accounts for the stiffness of the chain with $\kappa$ the bending rigidity and $\theta_i$ the local bending angle at monomer $i$. The second term represents the Pol II-Pol II interaction, where $E$ denotes the attractive interaction strength, and $f_{ij}$ equals 1 if monomers $i$ and $j$ occupy nearest neighboring sites on the lattice.

For simulations with a limited valency number, we defined an interaction list for each monomer with $s_i=1$. This list stores the genomic positions of the other monomers it interacts with and is constrained not to exceed the given valency number. It is updated after any polymer or TASEP moves.

Note that, due to the relatively high stall force of Pol II ($\sim$25-30 pN[106]), we assumed that Pol II-Pol II interactions do not impede Pol II elongation.

**Numerical simulations.** In our study, we set $\kappa \sim 1.2\, k_B T$ and a lattice volumic density of 50% to account for a chromatin fiber with a Kuhn length of 100 nm[69,107] and a typical base-pair density found in mammalian and fly genomes (~0.01 bp/nm³)[108]. Simulations were initiated by unknotted configurations[55] and performed with a kinetic Monte Carlo algorithm. In addition to the TASEP moves (see above), each MCS contains N local polymer trial moves. For each parameter set, 20 independent trajectories were conducted, discarding the first $10^6$ MCS from each trajectory to allow the system to reach a steady state. Subsequently, snapshots of the system were saved every $10^3$ MCS during the simulation during $10^7$ MCS and analyzed subsequently (see below). Since the characteristic spatial and time scales of the phenomenon under study (~100 nm, ~min) are well beyond the discretization scales (50 nm, 3 msec) imposed by the lattice and the kinetic Monte Carlo algorithm, the obtained results are not expected to depend qualitatively on the underlying modeling and simulation frameworks[109].

**Data analysis.** The radius of gyration (RG) provides a measure of the typical spatial extent of a gene, reflecting its overall span in 3D space. In a given configuration, the position of monomer $i$ can be defined as $\vec{r}_i \equiv (x_i, y_i, z_i)$. The RG is then calculated as follows:

$$RG = \sqrt{\frac{1}{n}\sum_{i=1}^{n}\left(\vec{r}_i - \vec{r}_m\right)^2}, \tag{5}$$

where $\vec{r}_m \equiv (x_m, y_m, z_m)$ is the mean position of all monomers.

To extract the diffusion coefficient ($D_i$) and exponent ($\alpha_i$) for monomer $i$, we first computed the time-averaged and ensemble-averaged mean-squared displacement, $<MSD_i>$, as a function of the time-lag, $\Delta t$. Subsequently, we performed a power-law fit of the form $D_i \Delta t^{\alpha_i}$ to the resulting curve using Numpy function numpy.polyfit($\log \Delta t, \log\langle MSD_i \rangle, 1$).

Furthermore, to establish a correspondence between simulation (MCS) and real (seconds) times, we compared our predictions with the typical MSD observed in yeast (~$0.01(\mu m^2/s^{0.5})\Delta t^{0.5}$, with $\Delta t$ in seconds)[76], leading to 1 MCS ~ 3 ms.

### Statistics and reproducibility
In Fig. 1B, C, genes smaller than 1 kb are excluded due to resolution limitations. In Fig. 2, clusters with fewer than 25 representative genes are excluded due to inadequate statistical significance. No additional statistical methods were applied to the analyses.

### Reporting summary
Further information on research design is available in the Nature Portfolio Reporting Summary linked to this article.

## Data availability
Processed data (intra-gene contact, RNA-seq and ChIP-seq enrichments, compartments and exon numbers for each gene > 1kbp) and source data of the figures are publicly available from Zenodo repository (Hossein Salari, 2024) at https://zenodo.org/records/10998192[110]. We also use publicly available data from Gene Expression Omnibus (GEO) through accession no : GSE130275, GSE178982, GSE90893, GSE90994, GSE16013, GSE85191, GSE195830 and ArrayExpress accession E-MTAB-9306.

## Code availability
Python notebooks for PMGA analysis and simulation codes are available on GitHub (https://github.com/physical-biology-of-chromatin/Transcription).

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

## Acknowledgements

The authors are grateful to Xavier Darzacq's lab for sharing processed data; Marco Di Stefano, Guillermo Orsi, and Aurèle Piazza for critical reading of the manuscript; Kerstin Bystricky, Tom Sexton, Giacomo Cavalli, Cédric Vaillant and the members of the Jost lab for fruitful discussions. We acknowledge Agence Nationale de la Recherche [ANR-18-CE12-0006-03, ANR-18-CE45-0022-01, ANR-21-CE45-0011-01] for funding. We thank PSMN (Pôle Scientifique de Modélisation Numérique) of the ENS de Lyon for computing resources.

## Author contributions

H.S. and D.J. designed the research; D.J. supervised the project; H.S. developed analytical tools and performed the research; H.S. and D.J. analyzed the data; G.F. provided conceptual advice; H.S. and D.J. wrote the paper with input from G.F.

## Competing interests

The authors declare no competing interests.
