## [Peer Review File · Nature Communications]

Transcription regulates the spatio-temporal dynamics of genes through micro-compartmentalizationReviewers' Comments:

Reviewer #1:

Remarks to the Author:

The role of active transcription and RNA PolII in organizing 3D genome structure is still under debate. Here the authors aim to study the role of transcription and PolII in the 3D genome, in particular gene structure, by combining publicly-available Micro-C data and biophysical modeling to generate quantitative models of the impact of transcription. The main claim of the paper is that transcriptional activity shapes 3D genome structure both at the level of local gene structure and global chromosome structure through PolII-mediated micro-compartmentalization.

The paper contains several interesting analyses and observations, but a major issue is that many of the conclusions and observations made in this paper have already been made in other studies, the authors just do not cite those studies (or more precisely, those studies are largely in the reference section, they are just not cited in the text when their claim is made again by the authors). Therefore, a required revision must be to comprehensively go through the entire manuscript and fully cite prior work, when conclusions from prior work is re-made in this paper.

The experimental observations that support the claim that active transcription compacts chromatin are mostly weak correlations from the reanalysis of available Micro-C data from mESCs. By quantifying PolII enrichment (IR) and contact enrichment (IC) for each gene, the authors claim that there is a 'significant positive correlation' between IC and IR scores. However, Fig. 1B shows at best a weak correlation between IC and IR, and the correlation is even lower when separating between A and B compartments in Fig. S3. Similarly, the fold changes upon TRP treatment in Fig 3 show almost no correlation, suggesting that the changes in PolII occupancy have little relationship to the changes in gene compaction. These results are almost identical to the analysis in Hsieh et al. (2020) where the data was generated, and do not help to resolve the conflict set out in the introduction about whether transcription compacts or decompacts genes.

Additionally, the gene length analysis in Fig. 2 lacks rationale and interpretation, and it is unclear how it contributes to the author's overall claims. For example, the authors investigated whether gene size could also be a determining factor, but it is not clear what gene size could be a determining factor for. As written, the authors describe a suite of analyses and observations relating to gene length, but what these correlations mean should be discussed.

The polymer simulations are interesting and the authors do a nice job of showing how the different parameters affect compaction. In Fig. 4H the parameters of the model were modified to better recapitulate PolII enrichment at the TSS and TTS, and the authors should show how IC changes with these new parameters.

For the polymer simulations, do they match the $P(s)$ vs. s^{-1} and MSD vs. $t^{0.5}$ determined from mESC Micro-C and mESC live-imaging studies, respectively? Fig. 6A suggest yes, but mESC data is not cited.

In Fig. 7, the authors claim that transcription mediates the formation of a phase-separated subcompartment between distal active genes. However, this claim is based on correlations, and a model that assumes attractive PolII interaction is almost bound to show interactions between gene bodies, especially for 100kb genes separated by 200kb (in contrast Fig. 7A shows genes half of the simulated length separated by more 800kb, more than 3x the distance). The claim about long-range subcompartments and transcription factories should be de-emphasized if this is the only line of evidence.

Minor comments:

- There is a typo in the x-axes of Fig. S4.
- The scale bars in the several figures (eg. Figs S8B, S9B, Fig 4F) do not appear to correspond to the

numbers in the heatmaps.

- Hi-C and Micro-C refer to different assays, but is used interchangeably several times (eg. Fig.7 legend says Hi-C but the main text refers to Micro-C).

- Ref 14 = Ref 57

Reviewer #2:

Remarks to the Author:

This is an interesting paper that probes how interactions with RNAP affects the conformations and possibly the dynamics of a transcribing gene. First, the topic is of great interest largely because of several interesting experiments. Second, there does not appear to be consensus on the mechanism of RNAP induced changes in chromatin dynamics. Towards this end, the current propose an interesting study. Before recommending publication, I want to raise some issue, which I hope the authors will consider. The players, not counting the many TFs that initiate transcription, are Cohesin (partner CTCF), RNAP, and chromatin.

1) The correlation between IC and IR in Fig. 1B is only 0.56, which is stated to be significant. Is there way to appreciate this statement? I ask because nominally this would not be considered to be that high. Besides, the spread is a lot.

2)The report that loop extrusion (LE) by cohesin plays a minor role in intra-gene compaction. The authors know that this is at variance with a few studies, the most recent one being Ref 47 - a combined experimental and simulation study. A few questions: (1) What is (are) the reasons(s) for the different conclusions? (2) Gene lengths are typically much less than the sizes of TADs that are supposedly generated by cohesin. Does the length mismatch explain the differences? (3) The authors of ref 47, if I understand correctly, also explain the experimental data with LE playing a critical role. I understand that in Ref 47 Hi-C data was used with presumably lower resolution than the Micro-C data used here. Could it be that the experiments are not sufficiently accurate to constrain the models?

3)Some technical questions. (1) The authors use lattice models, which are perfectly fine for computing universal characteristics. Is there a justification for its use here? In other words, had they used off-lattice models would they arrive at the same conclusions? (2) The TASEP model and the polymer model have certain parameters. First, it was unclear where the rates of RNAP binding unbinding etc come from. Also, the attraction value between the gene loci. Where does the value of the valence come from?

4) A model that supports the current finding that LE may not determine loci dynamics at the gene level was proposed based on the assumption that RNAP induces active forces (<https://doi.org/10.1101/2022.04.30.490180>). The consequences were in accord with Mayeshima experiments (transcription reduces loci mobility). Also, the snap shot in Fig. 5K (left) seems to show that gene (light blue) has some order. Is that the case? The authors might want to comment on this aspect.

Reviewer #3:

Remarks to the Author:

In this manuscript, Salari et al. uncover a generally positive correlation between intragenic chromatin interaction and Pol II occupancy at the gene by analyzing public Micro-C and ChIP-seq data. They further proposed a biophysical model to integrate the role of transcription dynamics within a polymer model of chromatin organization and demonstrated the close relationship between transcriptional activity and chromatin micro-compartmentalization. My concerns are elaborated on below.

Major concerns:

1. Could the authors distinguish between Ser-5P Pol II (typically associated with transcription initiation) and Ser-2P Pol II (commonly linked to transcription elongation) to assess the effects of their occupancy on intragenic compaction, given that genome-wide profiles for both are available in mESCs?
2. On page 14, the section titled "A transcription-associated subcompartment emerges from Pol II-mediated phase separation" is presented. However, this section lacks robust evidence to convincingly argue that Pol II contributes to forming the transcription-associated subcompartment through phase separation. More direct evidence is needed in this regard. Additionally, the relationship between chromatin compaction and phase separation is not clearly elucidated.

Minor concerns:

1. The manuscript would benefit from improved writing. For instance, using terms like 'ChIPseq' or 'RNAseq' is not official or standard usage.

Below is a point-by-point response to reviewers' concerns.

Comments from the reviewers are shown in Courier font.

Responses to the comments are shown in red.

Reviewer #1 (Remarks to the Author):

The role of active transcription and RNA PolIII in organizing 3D genome structure is still under debate. Here the authors aim to study the role of transcription and PolIII in the 3D genome, in particular gene structure, by combining publicly-available Micro-C data and biophysical modeling to generate quantitative models of the impact of transcription. The main claim of the paper is that transcriptional activity shapes 3D genome structure both at the level of local gene structure and global chromosome structure through PolIII-mediated micro-compartmentalization.

The paper contains several interesting analyses and observations, but a major issue is that many of the conclusions and observations made in this paper have already been made in other studies, the authors just do not cite those studies (or more precisely, those studies are largely in the reference section, they are just not cited in the text when their claim is made again by the authors). Therefore, a required revision must be to comprehensively go through the entire manuscript and fully cite prior work, when conclusions from prior work is re-made in this paper.

We thank the reviewer for her/his interest in our study and her/his comments that help us improve the manuscript.

Regarding the comment on the originality of our approach and on the citations of previous works:

- On the data analysis side, we fully agree with the Reviewer that some of our observations may overlap with previous works (in particular in the papers where the data we used were published) but using different - complementary - characterizations or having a finer analysis on some aspects. For example, the correlation between the enrichment in intra-gene contacts (IC) and RNA Pol II occupancy (IR) were already observed by Corces and Darzacq groups, with different observables. However, the non-monotonous dependency between IR and IC was not characterized, nor was the effect of gene length (or exon number).
- On the modeling side, our approach is very original and, to the best of our knowledge, no model has previously addressed the role of transcription dynamics in the folding of genes.

Following the reviewer's comment, we carefully read the manuscript and make sure we cite properly the prior research.

The experimental observations that support the claim that active transcription compacts chromatin are mostly weak correlations from the reanalysis of available Micro-C data from mESCs. By quantifying PolII enrichment (IR) and contact enrichment (IC) for each gene, the authors claim that there is a 'significant positive correlation' between IC and IR scores. However, Fig. 1B shows at best a weak correlation between IC and IR, and the correlation is even lower when separating between A and B compartments in Fig. S3.

The correlation analysis in Fig. 1B is conducted over 24,363 genes and the correlation is highly statistically significant (p -value $< 1e-200$, Line 105). In Fig. 1C, we further illustrate such a robust relationship by computing a correlation of 0.97 between the average IC and IR scores (L212), implying a consistent trend of increased gene condensation with higher RNA PolII occupancy *on average*. Nevertheless, as discussed in the Discussion (L432 & L515), the observed high heterogeneity indicates that specific genes may deviate from this average behavior, with some genes showing decompaction despite RNA Pol II occupancy. Our feeling is, on the contrary, that a global correlation > 0.5 is quite strong for such a comparison between two very different biological observables (IC & IR), knowing the expected biological heterogeneity.

As previously noted, the correlation within the B-compartment is comparatively much weaker than in the A compartment. This discrepancy suggests a nuanced interplay between Pol II-mediated condensation and heterochromatin, wherein the effects of Pol II activity may compete with the phase separation processes happening in the heterochromatic regions. This nuanced understanding is crucial for unraveling the complexities of the observed correlations and their implications for gene regulation. We add a sentence of that L123.

Similarly, the fold changes upon TRP treatment in Fig 3 show almost no correlation, suggesting that the changes in PolII occupancy have little relationship to the changes in gene compaction. These results are almost identical to the analysis in Hsieh et al. (2020) where the data was generated, and do not help to resolve the conflict set out in the introduction about whether transcription compacts or decompacts genes.

Following short-term TRP or FLV treatments, indeed, a mild correlation is observed between the loss in IC and loss in IR as already mentioned in Hsieh et al. (2020). However, as already noted at L154, although there are changes in Pol II occupancy on genes after these treatments, a substantial presence of Pol IIs persists. Actually, Fig. 3D demonstrates that even after re-grouping genes based on the remaining RNA PolIIs levels after treatment, the same correlation between averaged IC and IR scores is observed as in WT conditions. Therefore, the residual Pol II levels may explain the modest correlation observed in the fold changes during TRP treatment. Importantly, it means that the RNA Pol II level is predictive, in average, of the intra-gene contact levels whatever the conditions. We believe that this original finding (not mentioned in Hsieh et al) is, on the contrary, key and provides further evidence on the relation between Pol II occupancy and intra-gene condensation. We further insist on that in the revised manuscript (L162).

Additionally, the gene length analysis in Fig. 2 lacks rationale and interpretation, and it is unclear how it contributes to the author's overall claims. For example, the authors investigated whether gene size could also be a determining factor', but it is not clear what gene size could be a determining factor for. As written, the authors describe a suite of analyses and observations relating to gene length, but what these correlations mean should be discussed.

The rationale behind the gene length analysis is double: (1) as, in mammals, highly expressed genes are usually small and long genes, when active, are usually lowly expressed ; we wanted to investigate if our previous analysis of the relation between IR and IC was not influenced by gene length (already mentioned L126); (2) if the mechanism behind this relationship is related to phase-separation (as we demonstrated in the second part of our work), we expect an impact of gene length at similar Pol II density, as such transitions are cooperative. We add a sentence in the revised version (L131) to mention this.

The polymer simulations are interesting and the authors do a nice job of showing how the different parameters affect compaction.

We thank the reviewer for her/his enthusiasm.

In Fig. 4H the parameters of the model were modified to better recapitulate PolIII enrichment at the TSS and TTS, and the authors should show how IC changes with these new parameters.

Following the reviewer's comment, we investigate the effect of non-homogeneous RNA PolIII profiles on the average IC score. Our findings indicate a similar IC vs IR score (vs gene length) for homogeneous and non-homogeneous profiles. These results have been discussed in line 316. Additionally, we have included an extra figure (Fig.S20) in the supplementary material highlighting the impact of non-homogeneous RNA PolIII profile on IC score.

For the polymer simulations, do they match the $P(s)$ vs. s^{-1} and MSD vs. $t^{0.5}$ determined from mESC Micro-C and mESC live-imaging studies, respectively? Fig. 6A suggest yes, but mESC data is not cited.

In our simulations the exponent γ in $P(s) \sim s^{-\gamma}$ is ranging from -1.4 at low IR score to -1.1 at high IR score which are typical exponents observed in mESC Hi-C data. We decide not to mention that as we focus on the relative compaction (the observed-over-expected data) where the absolute effect of such an exponent is less important.

For the diffusion exponent α present in the $MSD \sim t^{\alpha}$ is 0.5, we found it independent of Pol II occupancy. We add a citation to mention that the predicted exponent is compatible with experimental live-imaging in mESC in L 374.

In Fig. 7, the authors claim that transcription mediates the formation of a phase-separated subcompartment between distal active genes. However, this claim is based on correlations, and a model that assumes attractive PolIII interaction is almost bound to show interactions between gene bodies, especially for 100kb genes separated by 200kb (in contrast Fig. 7A shows genes half of the simulated length separated by more 800kb, more than 3x the distance). The claim about long-range subcompartments and transcription factories should be de-emphasized if this is the only line of evidence.

In Fig. 7B, we present compelling evidence from experimental Micro-C data of significant long-range interactions among the gene bodies of highly active genes, with distances ranging from 128kb to 2000kb. Notably, such interactions are not observed for genes with low or moderate gene activities. We remove the term 'subcompartment' from the corresponding Results section (L388) to avoid confusion and replace it with terms related to the 'colocalization of' or 'long-range contacts between' active genes. However, we keep it in the Discussion to propose some interpretations.

In our modeling efforts, we now investigate several different situations with smaller and more distant genes. Fig. 7C has been updated to include the results of these simulations, demonstrating the model's capability to faithfully reproduce the observed long-range interactions among active genes.

Furthermore, our findings align also very well with recent studies on transcriptionally active subcompartments (Goel, V.Y., Huseyin, M.K. & Hansen, A.S. Nat Genet 55, 1048–1056 (2023)), which reported long-range interactions between active promoters and enhancers. Leveraging high-resolution Micro-C data, our study suggests that such long-range interactions may extend beyond promoters and enhancers to traverse the entire gene bodies. It underscores the importance of our investigation in expanding our understanding of genomic interactions, particularly in the context of active gene regions. We reformulate the corresponding text (L418-421).

Minor comments:

- There is a typo in the x-axes of Fig. S4.

The x-labels of Fig.S4 (Now Fig.S6) have been changed.

- The scale bars in the several figures (eg. Figs S8B, S9B, Fig 4F) do not appear to correspond to the numbers in the heatmaps.

In these figures the color bars are presented in a log2 scale, while the values are represented in a linear scale. We add an explanatory note in the captions of all relevant figures.

- Hi-C and Micro-C refer to different assays, but is used interchangeably several times (eg. Fig.7 legend says Hi-C but the main text refers to Micro-C).

We have addressed this inconsistency and all Hi-C and Micro-C experiments are now termed correctly.

- Ref 14 = Ref 57

We have addressed this issue.

Reviewer #2 (Remarks to the Author):

This is an interesting paper that probes how interactions with RNAP affects the conformations and possibly the dynamics of a transcribing gene. First, the topic is of great interest largely because of several interesting experiments. Second, there does not appear to be consensus on the mechanism of RNAP induced changes in chromatin dynamics. Towards this end, the current propose an interesting study. Before recommending publication, I want to raise some issue, which I hope the authors will consider.

We thank the reviewer for her/his enthusiasm and acknowledge her/him for her/his comments that greatly help us improve the manuscript.

The players, not counting the many TFs that initiate transcription, are Cohesin (partner CTCF), RNAP, and chromatin.

1) The correlation between IC and IR in Fig. 1B is only 0.56, which is stated to be significant. Is there way to appreciate this statement? I ask because nominally this would not be considered to be that high. Besides, the spread is a lot.

The correlation analysis in Fig. 1B is conducted over 24,363 genes and the correlation is highly statistically significant ($p\text{-value} < 1e\text{-}200$, L105). In Fig. 1C, we further illustrate such a robust relationship by computing a correlation of 0.97 between the average IC and IR scores (L212), implying a consistent trend of increased gene compaction with higher RNA PolII occupancy *on average*.

To complement this, and as a comparison, we computed the correlation between IC and repressive marks (H3K27me3 and H3K9me3), and observed much weaker (in amplitude) - but negative - correlations (see new Fig. S2 and corresponding statement in L112).

Regarding the spread, as already discussed in the Discussion (L432 & L515), the observed high heterogeneity indicates that specific genes may deviate from this average behavior, with some genes showing decompaction despite RNA PolII occupancy. Our overall feeling is that a global correlation > 0.5 is quite strong for such a comparison between two very different biological observables, knowing the expected biological heterogeneity.

2)The report that loop extrusion (LE) by cohesin plays a minor role in intra-gene compaction. The authors know that this is at variance with a few studies, the most recent one being Ref 47 - a combined experimental and simulation study. A few questions: (1) What is (are) the reasons(s) for the different conclusions?

While, indeed at a rapid glance, our results may seem contradictory from those Ref 47 (now Ref.42), they are actually quite consistent and complementary. In Ref 47, authors focus on the interplay between cohesin loop extrusion and transcription 'outside' genes, for example regarding the formation of TADs and showing that active genes may represent (more or less strong) barriers to loop extrusion. Our own investigation (pile-up plot around genes, Fig.3) similarly illustrates the significant impact of loop extrusion activity beyond gene regions, contrasting with its relatively minor role within the gene regions. The same observations can be done also (visually) in Fig.1&2 of Ref 47. We add a sentence in the Discussion on that (L439).

(2) Gene lengths are typically much less than the sizes of TADs that are supposedly generated by cohesin. Does the length mismatch explain the differences?

Most of the highly active genes are small. They are often found at TAD boundaries which is consistent with our pile-up analysis (Fig.2 and L145) and may actually play a role in TAD formation by being boundaries for loop extrusion (as suggested in Ref 47).

(3) The authors of ref 47, if I understand correctly, also explain the experimental data with LE playing a critical role. I understand that in Ref 47 Hi-C data was used with presumably lower resolution than the Micro-C data used here. Could it be that the experiments are not sufficiently accurate to constrain the models?

It is true that the resolution of the HiC map in Reference 47 does not permit a detailed examination of changes occurring specifically within the gene region. Despite this limitation, the concordance in our respective observations strengthens the consensus regarding the significant impact of loop extrusion activity, particularly in regions outside the genes.

3)Some technical questions. (1) The authors use lattice models, which are perfectly fine for computing universal characteristics. Is there a justification for its use here? In other words, had they used off-lattice models would they arrive at the same conclusions?

In the present work, our primary objective is to unravel the intricate interplay between transcription processes and genomic structure, which can be seen as 'universal' characteristics. Moreover, our group have used lattice models in the past extensively with a lot of success to study generic (eg, Tortora et al, PNAS 2023 ; Abdulla et al, Macromolecules 2023) but also specific (eg, Szabo et al, Sci Adv, 2020 ; Salari et al, Genome Res 2021) questions related to chromatin, with conclusions on the structural and dynamical sides that were consistent with similar works performed with MD when available. Other groups have also successfully employed such a lattice modeling to delve into localized questions (eg, Beagrie, Robert A., et al. Nature 543.7646 (2017), Messelink, Joris JB, et al. Nature communications 12.1 (2021)).

In addition, one advantage of our lattice implementation is that it is versatile (i.e. many different mechanisms can be easily implemented) while keeping a high computing efficiency. In our context, the integration of the TASEP model for transcription into the lattice model was technically much more straightforward than in standard MD engines.

Therefore, we did not try to use off-lattice or MD models for this question. As long as the characteristic scale of the phenomenon under study is beyond the discretization scale imposed by the lattice, which is the case here, we do not see any issue in using such an approach (see also Halverson, Jonathan D., Kurt Kremer, and Alexander Y. Grosberg. *Journal of Physics A: Mathematical and Theoretical* 46.6 (2013)). We add a sentence in the Materials and Methods regarding that matter (L614).

2) The TASEP model and the polymer model have certain parameters. First, it was unclear where the rates of RNAP binding unbinding etc come from. Also, the attraction value between the gene loci. Where does the value of the valence come from?

In the TASEP model, steady-state profiles of Pol II are dependent on the ratios of the various kinetic rates. We adjusted these ratios to faithfully replicate the typical average Pol II profiles observed experimentally. The remaining 'free' parameter is the elongation rate (γ) that determines the 'time-scale' in the TASEP simulations. We systematically investigated its impact to study the interplay between the transcription and the polymer/folding dynamics (Fig.4E) and then fixed it to biologically-relevant value (2kbp/min).

Regarding the interaction parameters, namely the strength of interaction and valency, which play pivotal roles, we systematically vary these parameters, striving to capture the nuanced features present in the experimental observations (Fig.4E,F, Fig.S19) and showing that our conclusions do not depend on their exact value.

4) A model that supports the current finding that LE may not determine loci dynamics at the gene level was proposed based on the assumption that RNAP induces active forces (<https://doi.org/10.1101/2022.04.30.490180>). The consequences were in accord with Mayeshima experiments (transcription reduces loci mobility). Also, the snap shot in Fig. 5K (left) seems to show that gene (light blue) has some order. Is that the case? The authors might want to comment on this aspect.

We thank the reviewer for this reference. We now cite it correctly in the revised manuscript (L466). The configuration (in Fig.5K) is mildly condensed and does not have a particular order (the confusion in the snapshot may have arisen from the lattice polymer model).

Reviewer #3 (Remarks to the Author):

In this manuscript, Salari et al. uncover a generally positive correlation between intragenic chromatin interaction and Pol II occupancy at the gene by analyzing public Micro-C and ChIP-seq data. They further proposed a biophysical model to integrate the role of transcription dynamics within a polymer model of chromatin organization and demonstrated the close relationship between transcriptional activity and chromatin micro-compartmentalization.

We acknowledge the reviewer for her/his comments that greatly help us improve the manuscript.

My concerns are elaborated on below.

Major concerns:

1. Could the authors distinguish between Ser-5P Pol II (typically associated with transcription initiation) and Ser-2P Pol II (commonly linked to transcription elongation) to assess the effects of their occupancy on intragenic compaction, given that genome-wide profiles for both are available in mESCs?

We have now computed the correlations between the intra-gene contact (IC) score and intra-gene RNA PolIII Ser 2P enrichment (as well as for 5P), see new Sup Fig.S3. Our analysis reveals a consistent correlation of IC with both RNA PolIII Ser 2P and 5P, emphasizing the robustness of this relationship across different phosphorylation states of RNA PolIII. Interestingly, average profiles of Ser2P do not have a peak at TSS and thus, Ser2p profiles alone cannot explain the average pattern observed in the pile-up analysis (PMGA). Ser 5p profiles alone are more consistent with PMGA (small peaks at TSS and TTS) but still less than total RNAPol II profiles which exhibit more pronounced peaks at TSS and TTS and are more consistent with the TSS-TTS loops and stripes observed in the data. We add sentences to discuss that (L108 and L141).

2. On page 14, the section titled "A transcription-associated subcompartment emerges from Pol II-mediated phase separation" is presented. However, this section lacks robust evidence to convincingly argue that Pol II contributes to forming the transcription-associated subcompartment through phase separation. More direct evidence is needed in this regard.

In Fig. 7B, we present compelling evidence from experimental Micro-C data of significant long-range interactions among the gene bodies of highly active genes, with distances ranging from 128kb to 2000kb. Notably, such interactions are not observed for genes with low or moderate activity levels. We remove the term 'subcompartment' from the corresponding Results section (L388) to avoid confusion and replace it with terms related to the 'colocalization of' or 'long-range

contacts between' active genes. However, we keep it in the Discussion to propose some interpretations.

Furthermore, our findings align also very well with recent studies on transcriptionally active subcompartments (Goel, V.Y., Huseyin, M.K. & Hansen, A.S. Nat Genet 55, 1048–1056 (2023)), which reported long-range interactions between active promoters and enhancers. Leveraging high-resolution Micro-C data, our study suggests that such long-range interactions may extend beyond promoters and enhancers to traverse the entire gene bodies. It underscores the importance of our investigation in expanding our understanding of genomic interactions, particularly in the context of active gene regions. We reformulate the corresponding text (L406).

Additionally, the relationship between chromatin compaction and phase separation is not clearly elucidated.

We are aware that the term 'chromatin compaction' may have several interpretations in the biological literature and is often used to describe the local fine structure of chromatin (nucleosomal scale or below). In our work, we refer to compaction more as a synonym of 'condensation' or 'folding', as we quantify it as an enrichment of contacts, thus scanning larger chromatin scales. In that context, the use of 'phase separation' is correct to discuss the spatial colocalization of active genes.

In the revised text, to avoid confusion, we decide to remove as much as possible the term 'compaction' and replace it with less confusing denominations.

Minor concerns:

1. The manuscript would benefit from improved writing. For instance, using terms like 'ChIPseq' or 'RNAseq' is not official or standard usage. We have carefully read again the manuscript and addressed all typos and non-official terms.

Reviewers' Comments:

Reviewer #1:

Remarks to the Author:

The authors have addressed most of the original concerns.

The only concern would be the section on "Transcription-associated long-range contacts between genes emerge from Pol II-mediated phase separation". There still does not seem to be evidence of a phase transition, the evidence only supports that PolII density correlates with inter-gene interactions, and the simulations suggest that PolII density is sufficient, but does not show necessity. The title is phrased to imply causality, which the evidence does not support.

Minor:

- It is not clear how IC and IR scores were calculated from the Materials and Methods.
- Fig 7B and 7C have missing tick mark labels.

Reviewer #2:

Remarks to the Author:

The authors have done a credible job of addressing my concerns. Therefore, I recommend publication.

Reviewer #3:

Remarks to the Author:

The authors have addressed our concerns. I recommend the publication of this work in Nature Communications.

Below is a point-by-point response to reviewers' concerns.

Comments from the reviewers are shown in Courier font.

Responses to the comments are shown in red.

Reviewer #1 (Remarks to the Author):

The authors have addressed most of the original concerns.

We appreciate the acknowledgment of the reviewer.

The only concern would be the section on "Transcription-associated long-range contacts between genes emerge from Pol II-mediated phase separation". There still does not seem to be evidence of a phase transition, the evidence only supports that PolIII density correlates with inter-gene interactions, and the simulations suggest that PolIII density is sufficient, but does not show necessity. The title is phrased to imply causality, which the evidence does not support.

Following the comment of the reviewer we change the title of the section "Transcription-associated long-range contacts between genes emerge from Pol II-mediated phase separation" to "Transcription-associated long-range contacts correlated with Pol II occupancy".

Minor comments:

- It is not clear how IC and IR scores were calculated from the Materials and Methods.

We add a new section into Methods to provide the details of calculation of IC and IR scores.

- Fig 7B and 7C have missing tick mark labels.

We add tick labels to figure 7B and 7C.

Reviewer #2 (Remarks to the Author):

The authors have done a credible job of addressing my concerns. Therefore, I recommend publication.

We thank the positive feedback and the acknowledgment of the reviewer. We sincerely appreciate his/her recommendation for publication of our manuscript.

Reviewer #3 (Remarks to the Author):

The authors have addressed our concerns. I recommend the publication of this work in Nature Communications.

We thank the positive feedback and the acknowledgment of the reviewer. We sincerely appreciate his/her recommendation for publication of our manuscript.